Your wish is my command! The influence of symbolic modelling on preschool children’s delay of gratification

Kumst S 1
Scarf D 2 damian@psy.otago.ac.nz
1 Faculty of Psychology and Neuroscience, Maastricht University , Maastricht , Netherlands
2 Department of Psychology, University of Otago , Dunedin , New Zealand
Deaner Robert
Electronic publication date: 2015 Feb 17
Publication date: 2015
Volume: 3
Electronic Location ID: e774
Received 2014 Dec 23; Accepted 2015 Jan 27
Copyright: © 2015 Kumst and Scarf
Copyright year: 2015
Copyright holder: Kumst and Scarf
License: This is an open access article distributed under the terms of the Creative Commons Attribution License, which permits unrestricted use, distribution, reproduction and adaptation in any medium and for any purpose provided that it is properly attributed. For attribution, the original author(s), title, publication source (PeerJ) and either DOI or URL of the article must be cited.
License URL: https://creativecommons.org/licenses/by/4.0/

Keywords: Delay of gratification, Symbolic modelling, Preschool children

Funding: Department of Psychology, University of Otago, Dunedin, New Zealand Funding was provided by the Department of Psychology, University of Otago, Dunedin, New Zealand. The funders had no role in study design, data collection and analysis, decision to publish, or preparation of the manuscript.

==============================
The ability of children to delay gratification is correlated with a range of positive outcomes in adulthood, showing the potential impact of helping young children increase their competence in this area. This study investigated the influence of symbolic models on the self-control of 3-year old children. Eighty-three children were randomly assigned to one of three modelling conditions: personal storytelling, impersonal storytelling, and control. Children were tested on the delay-of-gratification maintenance paradigm both before and after being exposed to a symbolic model or control condition. Repeated measures ANOVA revealed no significant differences between the two storytelling groups and the control group, indicating that the symbolic models did not influence children’s ability to delay gratification. A serendipitous finding showed a positive relationship between the ability of children to wait and their production and accurate use of temporal terms, which was more pronounced in girls than boys. This finding may be an indication that a higher temporal vocabulary is linked to a continuous representation of the self in time, facilitating a child’s representation of the future-self receiving a larger reward than what the present-self could receive.

Introduction

The ability to forgo an immediate reward and wait for a larger reward delayed in time is known as delay of gratification (Mischel & Underwood, 1974). In pre-school children, delay of gratification provides a simple measure of self-control and self-regulation. Interestingly, the ability of pre-school children to delay gratification is predictive of a range of life outcomes that extend well beyond the pre-school years. For example, pre-school children that are better able to cope with a delay cope better in stressful and frustrating situations during adolescence (Mischel, Shoda & Peake, 1988; Shoda, Mischel & Peake, 1990). Further, Casey and colleagues (2011) demonstrated that children who have difficulty delaying gratification at age 4 have difficulty resisting temptations and displaying cognitive control in adulthood. With respect to life outcomes, difficulty in delaying gratification in childhood is correlated with poorer cardiovascular, respiratory, dental and sexual health (Moffitt et al., 2011), a higher body mass index (Schlam et al., 2013), increased aggression (Ayduk et al., 2007), criminal offenses (Moffitt et al., 2011), substance abuse (Madden et al., 1997), and lower self-esteem, academic achievement (Li-Grining, 2007; Mischel, Shoda & Rodriguez, 1989; Wulfert et al., 2002) and social skills (McIntyre, Blacher & Baker, 2006) in adulthood. Given its predictive nature, improving the ability of children to delay gratification has a number of potential benefits.

Metcalfe & Mischel (1999) proposed a theoretical framework to explain why some children are capable of delaying gratification while others are not. In this framework, two distinct but interacting systems underlie the ability of children to delay. The “cool” or cognitive system is responsible for thought and reflection. This system allows the individual to reflect on the options at hand and select the one that best serves their long-term interests. The “hot” or emotional system is characterized by quick emotional processing and satisfying immediate needs. For example, in the delay-of-gratification maintenance paradigm (Mischel, Shoda & Rodriguez, 1989), the child is presented with an object they desire (e.g., a marshmallow), which is highly salient and activates the “hot” system, pushing children toward the immediately gratifying response of grabbing the marshmallow. On the other hand, the “cool” system allows children to reflect on the fact that, if they want to gain an even more desirable outcome (e.g., two marshmallows), they must wait. The balance between the “hot” and “cool” systems is thought to be an intrinsic trait, responsible for children’s differential performance on the delay-of-gratification tasks (Metcalfe & Mischel, 1999).

Importantly, however, numerous external factors can be used to either increase or decrease a child’s ability to delay gratification. For example, increasing a child’s attention to the salient attributes of the immediate reward, such as looking directly at the reward or talking about it, results in shorter wait times (Mauro & Harris, 2000; Mischel & Underwood, 1974; Putnam, Spritz & Stifter, 2002), while distraction techniques such as covering up the reward or thinking of the reward in a non-consummatory fashion results in longer waiting times (Sethi et al., 2000). In addition to salience and attentional strategies, parental style and attachment also predict delay behaviour. The children of parents who have an authoritarian parenting style tend to delay gratification while those with permissive parents do not (Mauro & Harris, 2000). Maternal regulatory strategies, such as distraction, reasoning, bargaining, indirect and direct commands, also impact children’s performance, with distraction being the most effective strategy (Putnam, Spritz & Stifter, 2002).

Empirical research on the strategies and parenting styles that contribute to delay of gratification has been used to propose guidelines to improve self-control in childhood (Strayhorn Jr, 2002). One of these guidelines suggests modelling and fantasy rehearsal. Bandura & McClelland’s (1977) social learning theory posits that behaviour is learnt from the environment through observation. Models (e.g., parents) explicitly convey how tasks should be performed and, as a result, shorten how long it takes children to learn a task or skill. The model may be a live individual demonstrating a behaviour, a verbal description or explanation of a behaviour, or symbolic (i.e., the behaviour is modelled in a book, on television, or other media) (Bandura & McClelland, 1977). A number of factors contribute to the influence of models on children’s behaviour. For example, the more the child identifies with the model, the more likely they are to reproduce the behaviour (Bandura & McClelland, 1977).

Bandura & Mischel (1965) demonstrated the influence models have on the performance of 4- and 5-year-olds on the delay-of-gratification task. After being tested on the task, children either observed an adult enact the opposite decision to what the child had just made or listened to a verbal description of the behaviour. For example, if the child chose to wait, the adult did not wait. Following this manipulation, children were immediately re-tested and again after a 4- to 5-week delay. Tellingly, children who at first delayed gratification shifted to choosing the immediate reward and children who initially chose the immediate reward tended to favour waiting for the delayed reward. Of note is the fact that the live model and the verbal model had a comparable impact on the immediate test, but only the live model influenced children’s behaviour at the 4- to 5-week follow-up.

An attribute of symbolic models is that they can take the form of a story, which is attractive for young children who regularly have storybooks read to them. Illustrated stories are particularly effective with preschool children because they are visually interesting and easy to memorize (Strayhorn Jr, 1988). Picture books have the potential to provide children with information about the world, allowing them to acquire additional knowledge to what they learn by personal experience (Heath, Houston-Price & Kennedy, 2014). For example, picture books influence a child’s healthy eating behaviour (de Droog, Buijzen & Valkenburg, 2014; Heath, Houston-Price & Kennedy, 2014) and facilitate the acquisition of novel words and their extension to real objects, especially when these objects are realistically depicted in the picture book (Ganea, Pickard & DeLoache, 2008; Tare et al., 2010). Further, 18- to 30-month-old children can learn simple novel actions through picture book readings (Simcock & DeLoache, 2006).

In addition to the pictures displayed, the behaviour of characters or the comments they make can also be used to show children how to achieve specific goals or acquire coping strategies (Strayhorn Jr, 1988). Narrations in storybooks may provide information about resolving problems successfully and indicate the personal qualities and behaviours that are associated with the achievement of a goal. Therefore, storybooks may foster messages coming directly from caregivers about which beliefs and behaviours are fundamental for success (Suprawati, Anggoro & Bukatko, 2014). Through repeating the story, as children often like to do, the child is free to imaginatively experience the behaviour or cognitive pattern of the model and rehearse it. Repetition may also help with abstraction, aiding a child’s ability to transfer the behaviour to situations beyond those conveyed in the story (Strayhorn Jr, 1988). It is known that the effectiveness of modelling can be promoted by presenting a model who encounters similar difficulties to the child, and who models how to overcome these difficulties through self-directed instruction (Meichenbaum, 1971). Children who have trouble in activities similar to those the model experiences are more likely to imitate the model than children who are successful in those activities (Gelfand, 1962), and models whose behaviour is rewarded are more likely to be imitated than models whose behaviour is punished (Bandura, 1965).

Although the potential influence of symbolic models has been discussed extensively in the literature, few studies have actually investigated the potential impact of symbolic models on the ability of children to delay gratification. In one of the few studies, Lee et al. (2008) investigated whether the ability of preschool children to delay gratification could be influenced by either explicitly labelling the children as being very patient individuals, saying that they had heard they were able to concentrate well and do boring things for a long period of time, or by reading a story to them in which an impulsive child received only one gift whereas a patient child received two gifts. The children who were labelled as patient delayed significantly longer than children in the control group, demonstrating the power of self-perception and priming. However, no statistically significant differences were found between the storytelling group and the control group, although they did delay one minute longer on average than controls (Lee et al., 2008). The limited effect of storytelling in this study could be due to the fact that the story was only read once, and to a full classroom. It is an open question as to whether reading the story to children individually, and on multiple occasions, would produce more promising results.

The current study was designed to examine the effects of both personal and impersonal storytelling on a 3-year old’s ability to delay gratification. Children were given storybooks to take home included either a personal or impersonal story, and parents were asked to read the storybooks to their children for approximately 1 week. Critically, the child’s ability to delay gratification was assessed both before and after the storybook intervention. We hypothesized that that children in the personal and impersonal storytelling conditions would delay gratification longer than those in the control condition, because models in the story demonstrated that waiting will be rewarded with a large gift and being impatient leads to a small gift. Further, it was predicted that children in the personal storytelling condition, whose names were included in the story, would wait longer than children in the impersonal storytelling condition. This second prediction was based on previous findings demonstrating that perceiving oneself as similar to the model enhances reproduction of the modelled behaviour and that observing similar others succeed reinforces one’s belief in one’s own capabilities (Bandura & McClelland, 1977; Schunk, 1987).

Method

Participants

A total of 83 (37 boys and 50 girls) 3-year-old children participated in the present study (Mean age: 34 months 20 days, Range: 30 months 17 days to 39 months 27 days). Three additional parents withdrew their children from the study after participating in the pre-test due to family reasons. One child was excluded from the sample because their post-intervention test took place one month after the pre-intervention test, due to unforeseen delays. All children were recruited from the Early Learning Project database of parents who had expressed interest in taking part in research at the University of Otago and participated with written consent from their parents. At the end of the experimental session, children received a small gift and parents received a patrol voucher to compensate them for their travel costs to and from the laboratory. This study was approved by the Human Ethics Committee at the University of Otago (Approval Number: 11/106).

Materials

The experimental room contained a sofa and a child’s table with four different coloured chairs. Parents were asked to remain seated on the sofa behind the child, who sat at the table facing away from them. On the table in front of the child were two closed gift boxes of different sizes: the small gift box measured 8 cm in width and 6 cm in height, and the large gift box 16 cm in width and 13 cm in height. Each gift box could be opened by lifting the lid, to which an ornamental ribbon was attached. Gift boxes were chosen over wrapped gifts because taking a peak under a lid is easier than unwrapping a gift, presumably making it more tempting for children. The small gift box contained five stickers, and the large gift box contained either a male or female Playmobil® figure.

Six versions of the picture book were created. The images in the picture book were the same for all conditions, except that girls saw princesses and fairy godmothers, while boys saw princes and wizards. Only the content of the story changed from one condition to the next. In the control story, two princes or princesses go for a walk in the forest, where they meet a benevolent magical figure (wizard or fairy godmother). One child is a bit hungry and is given a lollipop, while the other child receives a toy that they had supposedly lost in the forest and which the magical figure had found. In contrast to the control story, the personal- and impersonal modelling stories deliberately model a delay-of-gratification scenario. In these stories, the princes or princesses come across the magical figure and are offered a choice between receiving a small gift immediately or a larger gift later. One child is impatient while the other is willing to wait. The impatient child only receives a lollipop while the patient child receives a teddy bear (girls) or toy car (boys) after waiting for the magical figure to return with more ‘present potion’ or ‘magic present dust.’ The only difference between the personal- and impersonal modelling stories was that the patient prince or princess was named after the participant in the personal condition. All parents were asked to read the assigned storybook to their child every night before bedtime for 1 week.

Procedure

Families were contacted by telephone and were given a description of the experiment. When parents agreed for their child to participate in the study, the child was randomly assigned to one of three conditions: personal storytelling (n = 27), impersonal storytelling (n = 27) and control storytelling (n = 29).

Behavioural measures

Pre-intervention assessment

In the initial phase of the experiment, the children’s ability to delay gratification was assessed using the maintenance paradigm (Mischel, 2014; Mischel, Shoda & Rodriguez, 1989). This paradigm consists of a single trial where the total amount of time that the child is willing to wait for a reward is measured.

Children were tested individually in a quiet room by a female experimenter. The experimenter brought the participant to the laboratory where the child was given 5 min to get accustomed to the room and its decorations. During this time the parent read and signed the consent form and completed a questionnaire. The experimenter then seated the child at the table, facing away from their parent. The behaviour of the children was filmed by a camera that was set up in the room. Parents stayed in the room with the child during the experiment and were asked not to distract the child by asking questions, directing the child’s attention to things in the environment, or making comments concerning the task.

The children were shown both gift boxes, and were told that if they did not peek into the little gift box until the experimenter came back into the room (15 min) that they could keep what was in the little gift and would additionally receive what was in the larger gift box. The time started when the experimenter left the room, taking the larger gift with them. The dependent measure was the number of minutes between the moment the experimenter closed the door and the moment the child opened the lid of the gift box or the end of the specified 15 min period. Upon completion of the task, the children were given their assigned storybook and parents received instructions to read the book to their children once a day for 1 week.

Post-intervention assessment

One week later, all children returned to the laboratory for a second test of their ability to delay gratification. The same procedure was used as in the pre-intervention.

Cognitive measures

Parents filled out a questionnaire created by Busby Grant & Suddendorf (2011) assessing their child’s production and accurate use of temporal terms. The questionnaire is composed of a list of 18 temporal terms and phrases. Parents are asked to indicate whether their child uses these terms by answering with yes or no, and to rate how frequently and accurately their child uses these terms on a five-point Likert scale (1 = ‘never’, 2 = ‘occasionally,’ 3 = ‘sometimes,’ 4 = ‘often,’ 5 = ‘always’). Parents completed the questionnaire prior to children being assessed.

Results

The performance of children in each condition is shown in Table 1. Across the three groups, 30/83 children (Personal story: 8/27, Impersonal story: 10/27, Control: 12/29) waited the full 15 min at pre-test. With these children included, the average time children were willing to wait at the pre-intervention assessment was 7.81 min (Personal story: 7.94 min., Impersonal story: 7.18 min, Control: 8.29 min). Importantly, the average time children were willing to wait did not differ across the three conditions pre-intervention, F(2, 80) = .218, p = .804.

Table 1 The performance of all children on the delay-of-gratification task.

The performance of all children in the personal story, simple story, and control story conditions.

Group	Personal	Impersonal	Control	Boys	Girls	
Pre-test						
Mean	7.94	7.18	8.29	6.42	8.94	
SD	5.94	6.78	6.39	5.93	6.46	
N (boy/girl)	27 (12/15)	27 (12/15)	29 (13/16)	37	46	
Post-test						
Mean	8.21	8.03	7.96	6.15	9.60	
SD	6.64	6.94	6.71	6.71	6.32	
N (boy/girl)	27 (12/15)	27 (12/15)	29 (13/16)	37	46	

To assess the potential impact of symbolic modelling, we conducted a repeated-measures Analysis of Variance (ANOVA) with Condition (2: Personal storytelling vs. Impersonal storytelling vs. Control) and Gender (2: girls vs. Boys) as factors. The analysis revealed no significant effect of Session, F(1, 76) = .890, p = .348, suggesting the length of time children were willing to wait did not differ between the pre- and post-intervention phases. Further, there was no Session by Condition interaction, F(2, 76) = .316, p = .730, suggesting the absence of any change between pre- and post-intervention was true for all three conditions. Finally, there was no Session by Condition by Gender interaction, F(2, 76) = .104, p = .902, suggesting the absence of any change between pre- and post-intervention, for any condition, was true for both boys and girls. Overall, this shows that the modelled behaviour in the fairy tales did not lead to a statistically significant improvement in the children’s self-control.

The absence of any change between the pre- and post-intervention sessions could be due to a ceiling effect as 30 (36%) children waited the 15 min at the pre-intervention assessment and consequently could not further improve at the post-intervention assessment. Therefore, a second repeated-measures ANOVA was conducted with these children excluded. A significant effect of Session was found, F(1, 47) = 4.29, p = .044, suggesting a change in the length of time children were willing to wait between pre- and post-intervention, but no interaction was found between Session and Condition, F(2, 47) = .686, p = .509, indicating that this change between pre- and post-intervention was present in all conditions. Finally, no Session by Gender by Condition interaction was found, F(2, 47) = 13.64, p = .452, suggesting the change between pre- and post-intervention was observed in all conditions, for both girls and boys. The fact that children were willing to wait longer in the post-intervention assessment may be due to the children being more comfortable with the experimenter and the context of the experiment. Since this improvement in waiting time was observed across all conditions we can once again conclude that symbolic modelling did not influence the children’s ability to delay gratification (Table 2).

Table 2 The performance of children that did not wait the full 15 min pre-intervention.

The performance of children in the personal story, simple story, and control story conditions that did not wait the full 15 min at the pre-intervention test.

Group	Personal	Impersonal	Control	Boys	Girls	
Pre-test						
Mean	4.97	2.58	3.55	4.05	3.38	
SD	4.42	3.76	3.74	4.28	3.81	
N (boy/girl)	19 (10/9)	17 (9/8)	17 (10/7)	29	24	
Post-test						
Mean	6.11	5.42	4.29	4.15	6.70	
SD	6.16	6.74	5.86	5.79	6.47	
N (boy/girl)	19 (10/9)	17 (9/8)	17 (10/7)	29	24	

Out of exploratory interest we investigated whether the children’s use of temporal terms was related to their ability to delay gratification. We correlated the children’s mean waiting time (across pre- and post-intervention assessments) with scores obtained on the three questions in the temporal questionnaire. The number of temporal terms used by the children was significantly correlated with their ability to delay gratification, n = 83, r = .259, p = .018. Further, the frequency with which children used these terms, n = 83, r = .303, p = .005, and how accurately they used them, n = 83, r = .335, p = .002, also correlated with their waiting time. Importantly, these effects held when Age was partialed out (use of temporal terms, n = 83, r = .242, p = .029, frequency, n = 83, r = .287, p = .009, and accuracy, n = 83, r = .322, p = .003) suggesting this effect is not simply due to the fact that older children likely use these terms more frequently and have longer waiting times.

With respect to gender, there was a significant effect of gender on the time children were willing to wait, F(1, 76) = 4.825, p = .031, but no Gender by Time interaction, F(1, 76) = .355, p = .553, suggesting this effect was consistent across the pre- and post-intervention assessments (Table 1). On average, pre-intervention girls waited 8.94 min and boys waited 6.42 min, with a similar difference post-intervention, where girls waited 9.60 min and boys waited 6.15 min. To investigate whether this difference was related to the children’s use of temporal terms, we conducted a series of independent-sample t-tests comparing boys and girls across the three time questionnaire measures. Interestingly, compared to boys, girls used significantly more temporal terms, t(81) = 2.912, p = .005, d = .63, and used them both more frequently, t(81) = 2.636, p = .010, d = .58, and more accurately, t(81) = 2.243, p = .028, d = .49 (Fig. 1).

Figure 1 Questionaire and behavoural measures.

Differences between girls and boys in (A) the use of temporal terms; (B) the frequency with which they use temporal terms; (C) the accuracy of use of temporal terms; (D) the amount of time they were willing to wait across pre- and post-intervention. The error bars represent the standard error of the mean.

Discussion

The aim of the current study was to investigate whether the self-control of 3-year-olds could be improved by exposing them to a symbolic model demonstrating self-control. It was hypothesised that children would assimilate the behaviour of the patient child in the storybook and, as a result, be willing to wait longer for a large reward. Further, it was hypothesised that naming the symbolic model after the child (personal storytelling condition) would have a greater influence on the children’s wait times than a symbolic model with an unrelated name (impersonal storytelling condition). In contrast to our predictions, neither children in the personal nor impersonal storytelling condition waited longer at the post-intervention assessment, and the wait times of children in these two conditions were no different to those of children in the control group. When children who waited the full 15 min at the pre-intervention assessment were excluded, a significant difference was found between pre- and post-intervention assessments but this difference was not influenced by condition. It must be acknowledged, however, that excluding the children who waited the 15 min on the pre-intervention assessment reduces the statistical power of the analysis and thus we must interpret the absence of an effect of condition with caution. The improvement between the pre- and post-intervention assessments may be due to a practice effect, with the presentation of the same task at pre- and post-intervention assessments resulting in the child being more familiar with the procedure and the task, and thereby enhancing his or her performance (Schmidt & Teti, 2006). Also, in the post-intervention phase children may have (correctly) assumed that the content of the gift box sitting in front of them was the same as in the pre-intervention phase, perhaps making it less tempting to open.

With respect to symbolic models, our storybooks may not have influenced the children’s self-control because, being between 2 and 3 years of age, the children in the current study may not yet have developed the conceptual understanding required to assimilate the behaviour. Although the children in the current study could likely understand the story, they may not have understood that the story demonstrated the concept of patience and therefore did not apply this concept to their own behaviour. Sloutsky (2003) suggests that children learn concepts through both perceptual and attentional mechanisms, detecting similarities or discrepancies in the environment and creating categories based on the correspondences they observe. Presenting examples of the concept and contrasting examples helps children detect the important features and facilitates categorization. Concrete concepts (e.g. book, animal, tree) are learnt more easily than abstract concepts (e.g. patience, fairness), because perceptual similarities facilitate the categorization of these objects. Little is known about the development of abstract concepts (Baddeley, 1999; Sloutsky, 2003). It is possible that young children must be exposed to abstract concepts multiple times in varying contexts to fully understand the meaning of these concepts and to apply them in their everyday life. Further, Simcock & DeLoache (2006) showed that the understanding of the relation between a symbol and what it stands for is in its early stages at 2.5 years, supporting the statement that the children in our study may not have benefited from our picture-book due to their still-developing conceptual understanding.

Separate from the concept of patience, our results suggest that the children’s knowledge of the benefits of waiting (i.e., receiving a bigger present) was not sufficient to keep them from opening the present before the 15 min elapsed. This discrepancy between knowledge and behaviour has also been demonstrated on card-sorting tasks where children must switch between sorting rules. Zelazo, Frye & Rapus (1996) presented children with a set of cards that had coloured shapes on them and had the children sort them according to a single rule (e.g., by colour). After learning the first rule, children were taught a new rule and were asked again to sort the cards by this new rule. Interestingly, 3-year-old children expressed knowledge of the new rule, but continued to use the old rule, whereas 4-year-old children displayed no difficulty switching between the two rules. Zelazo, Frye & Rapus’s (1996) findings demonstrate that, for children aged 3 years and younger, knowledge is not always sufficient to influence actions. In the current study, it is possible that the children’s knowledge of the benefits of waiting was not sufficient to overcome their prepotent response to open the gift. It is possible that children aged four and older may benefit more from symbolic models in storybooks than younger children.

In our storybook, both the non-patient and the patient child received a gift from the magical figure: the first received a lollipop and the second received a toy. Social learning theory suggests that the rewards associated with the modelled behaviour influence the likelihood of the behaviour being imitated (Bandura & McClelland, 1977). It is possible that some children perceived the lollipop as more desirable than the toy and, consequently, displayed similar non-patient behaviour on our laboratory task. To exclude this possibility, it would be interesting to vary the magnitude (e.g., size), rather than the type, of the reward the child receives if they wait. Bandura & McClelland’s (1977) theory also recognizes the importance of perceived similarity between the model and the learner, which may have been lacking in our storybook due the stories being more fantasy-based. Placing pictures of actual children in the storybook would improve the similarity between the book and the real world and may increase children’s self-efficacy (Dowrick, 2000).

A serendipitous finding of the current study was the relationship between the children’s production and accurate use of temporal terms and their ability to wait for the larger reward; that is, the more temporal terms children used and the more accurately they used them, the longer they were willing to wait. This effect held when age was partialed out, suggesting it was not simply due to older children having both a better vocabulary and longer waiting times. Previous studies have shown that children’s use of temporal terms begins at around the age of 2 and rapidly grows thereafter. Busby Grant & Suddendorf (2011) suggested that 3-year-olds often and accurately use terms representing the present and broad temporal terms such as ‘soon’ or ‘later.’ Specific terms related to the future or the past (e.g., yesterday, tomorrow) are learnt later than general terms (e.g., when I was little), between the age of 3 and 5 years. In our study, girls had a larger temporal vocabulary than boys and, interestingly, also waited longer than boys. Surprisingly, Busby Grant & Suddendorf (2011) did not find gender differences in the use or accuracy of temporal terms.

Consistent with our findings, a recent meta-analysis reported that girls delay gratification for longer periods of time (Silverman, 2003). Speculatively, our finding suggests this link may be due to the ability of girls to use temporal terms, and that their use of temporal terms may reflect a clearer understanding of the past and future. It is possible that children with a larger temporal vocabulary are better able to project themselves into the future, and are willing to delay gratification to give their future-self a larger reward than what the present-self would receive. It is through conversations about the self in noncurrent situations that children learn about their extended self in time (Moore & Lemmon, 2001; Nelson, 2001). Between the ages of 2 and 5, children acquire the sense of self as continuous in time, with increasing capacity to talk about different experiences (Nelson, 2001), and it is also between these years that the ability of children to delay gratification slightly increases (Lemmon & Moore, 2007). To our knowledge, no studies have analysed the relation between knowledge of temporal terms and the ability to sacrifice an immediate reward in favour of a future larger reward. It is possible that the more specific temporal terms children understand and use, the better their understanding of time and the better their representation of the self in time.

In our study, we did not measure the children’s general language competence which may also relate to their ability to delay gratification. Language impairments are known to correlate with maladaptive outcomes, such as ADHD (Bruce, Thernlund & Nettelbladt, 2006; Cohen et al., 2000), aggressive behaviours (Dionne et al., 2003; Estrem, 2005) and poor problem solving (Baldo et al., 2005; Cohen et al., 1998), which are all linked to low self-control. Language skills are thought to enhance the self-regulatory competence of children by enabling them to express themselves verbally instead of emotionally, reflect on their behaviour, and occupy themselves in the presence of emotional circumstances (Cole, Armstrong & Pemberton, 2010). For example, language may allow children to reflect on rules and shift their attention, rather than focusing on items that they cannot have (Roben, Cole & Armstrong, 2013). To our knowledge, very few studies have analysed the link between language and self-control. Beaver et al. (2008) conducted a longitudinal study from the beginning of kindergarten to the end of first grade over four time points. They discovered that language skills significantly correlated with self-control at the beginning of kindergarten; this result held over years, showing that higher language competences are associated with higher self-control. In their longitudinal study following children from 18 to 48 months of age over four time points, Cole, Armstrong & Pemberton (2010) analysed how early language development influenced anger expressions and the use of regulatory strategies. They found that language skills and the rate of language growth at 24 and 36 months were a strong predictor of anger expressions at 48 months, with higher language skills being associated with lower anger expressions. Higher language skills were also associated with children’s initiation of seeking the support of their mothers at 36 and 48 months, which lead to less angry expressions. The language skills of children were also associated with their ability to distract themselves at 36 and 48 months of age, indicating that higher language skills are associated with a higher frustration threshold. These two studies show a significant link between self-control and language competence. Therefore, in the future it would be of interest to study the relationship between children’s general language competences, their understanding of time and their ability to delay gratification, in order to identify how each one contributes to variations in self-control.

With respect to limitations, in the present study we did not assess the children’s understanding of the theme or message of the storybook, although it is important to note that the book was short, had a simple story line, and was read to the children on multiple occasions. It would be beneficial in future studies to assess the children’s understanding of the book, perhaps by using the storybook as a prop and asking children to show and tell the experimenter what happens in the book. Previous studies have demonstrated that this show and tell technique is effective with young children (Hayne & Imuta, 2011; Scarf et al., 2013). A second limitation is that parents, and often siblings, were in the experimental room during the delay-of-gratification task. This may have influenced the children’s behaviour by distracting them or by giving implicit signals to not open the present (e.g., movements of the head, moving when the child reached for the gift) (Bandura, 1992; Putnam, Spritz & Stifter, 2002). Finally, parents may have coached their child between the pre- and post-intervention tests. For example, one parent noted that they tried to train their child on a delay-of-gratification task at home and thoroughly discussed the behaviours described in the book.

In summary, the present suggests 3-year-old children do not assimilate a behaviour modelled in a storybook into their own behaviour. Between pre-school and primary school, a child’s language ability and conceptual understanding increases dramatically and, therefore, symbolic models may be more effective for children slightly older than those used in the present experiment. The positive relationship observed between the children’s temporal vocabulary and their ability to delay gratification suggests a larger temporal vocabulary may be linked to children’s understanding of the future and the past, and this may make it easier for children to envisage the self in the future receiving the large reward, making the small reward less tempting. Future studies may look to investigate the possible relationship between the ability to delay gratification and children’s perception of the extended self in time.

Supplemental Information

Supplemental Information 1 Data set with participants names removed

Click here for additional data file.

Additional Information and Declarations

Competing Interests

Author Contributions

Human Ethics

The authors declare there are no competing interests.

S Kumst conceived and designed the experiments, performed the experiments, wrote the paper.

D Scarf conceived and designed the experiments, analyzed the data, contributed reagents/materials/analysis tools, prepared figures and/or tables, reviewed drafts of the paper.

The following information was supplied relating to ethical approvals (i.e., approving body and any reference numbers):

This study was reviewed and approved by the University of Otago Human Ethics Committee (Approval number 11/106).

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
