# Peer review of "Your wish is my command! The influence of symbolic modelling on preschool children’s delay of gratification"

_PeerJ, doi:10.7717/peerj.774_

## Round 0.1 · original submission · Minor Revisions

· Academic Editor

Minor Revisions

Dear Dr. Kumst and Dr. Scarf,

I have received reviews from two experts regarding your submission.

Review #1 (DeLisi) reported no problems and recommended Acceptance.

Review #2 (Anonymous) identified minor issues that should you should address. They recommended Minor Revision.

I have also read the manuscript carefully. I found no major problems, but there are several minor issues should you should address. These are described below.

Therefore, my decision is Minor Revision.

If you decide to revise the work, be sure to submit a detailed list of changes or a rebuttal against each point raised by the reviewer or myself when you submit the revised manuscript.

Thanks again for considering PeerJ as an outlet for your research. We hope you will continue to do so.

Sincerely,
Robert O. Deaner



Line 177: Please provide evidence or at least some references indicating that this “maintenance paradigm” is a conventional, valid delay of gratification task.

Line 198: Please indicate when parents completed the cognitive measures questionnaire. Ideally, this should have been completed prior to children doing the delay of gratification task, both pre-test and post-test. If this was the case, make this clear. If this wasn’t this case, make this clear AND acknowledge this potential weakness in your Discussion (probably around line 360). The potential problem, of course, is circularity: parents may base their responses on children’s performance in the task.

Line 252: Please indicate what the error bars in Figure 1 indicate. Presumably they are standard errors, but this should be made clear.

Line 256: Somewhere in the Discussion (probably around line 360), please acknowledge and briefly discuss the limitations of your conclusion with respect to the study’s limited statistical power. For example, once the children who initially reached the ceiling were eliminated, you only had 17 children in the control group. It should be acknowledged that having a larger sample would allow readers have more confidence in your negative conclusion. A power analysis may be worth conducting and discussing.

Line 364: Please acknowledge and briefly discuss whether you obtained any data (or have some ideas) about whether children actually understood the key events in the story. This would seem like something that could have been assessed; if it wasn’t, it should be acknowledged as a potential weakness in the study.

·

Basic reporting

The basic reporting is fine

Experimental design

I have no methodological concerns

Validity of the findings

The author does not overstate the findings and acknowledges limitations.

Additional comments

I like the study, and appreciate the use of 3 years olds that moves beyond the 4 year olds of Mischel's work. It is important to note the emergence of self-control and its intersections with externalizing behaviors. The author might be interest in Mischel's new book, The Marshmallow Test. It connects well with the current study.

Reviewer 2 ·

Basic reporting

This manuscript reports on a study aiming to assess whether symbolic modelling influences 3-year-old children ability to delay gratification. Eighty-three children were tested in a delay maintenance task both before and after a one-week treatment in which their parents were required to read them a book (i.e., the symbolic modelling manipulation). Children were randomly assigned to one of three conditions, differing for the content of the book: (i) personal story-telling, (ii) impersonal story-telling, and (iii) control. It emerged no significant effect of the condition, but an interesting correlation between children’s ability to delay gratification and their use of temporal terms, as assessed by a questionnaire filled out by the parents.

This is a very well written paper and I have only minor comments:

- l 44: “…(Moffit et al. 2011), substance abuse (Madden et al. 1997), lower self-esteem…”
- l 45: please replace the semicolon after “Wulfert 2002” with a comma
- l 47: “proposes”
- l 62: “longer waiting times”
- l 95: “…the behaviour of characters or the comments…”
- l 125: “We hypothesized that the children…”

Experimental design

- ll 236-240: please include N for the correlations
- l 243: “Gender by Session interaction” (for consistency with ll 223-226)

Validity of the findings

- l 296: “…for children aged 3 years…”

---

## Round 0.2 · accepted · Accept

· Academic Editor

Accept

This is an interesting, high-quality study. It should make a nice contribution to the literature.